# Utility of Lean Body Mass Equations and Body Mass Index for Predicting Outcomes in Critically Ill Adults with Sepsis: A Retrospective Study

**DOI:** 10.3390/diseases12020030

**Published:** 2024-01-26

**Authors:** Rumiko Shimizu, Nobuto Nakanishi, Manabu Ishihara, Jun Oto, Joji Kotani

**Affiliations:** 1Division of Clinical Pharmacy, Faculty of Pharmaceutical Sciences, Kobe Gakuin University, 1-1-3 Minatojima, Chuo-ward, Kobe 650-8586, Japan; simizu@pharm.kobegakuin.ac.jp; 2Division of Disaster and Emergency Medicine, Department of Surgery Related, Kobe University Graduate School of Medicine, 7-5-2 Kusunoki, Chuo-Ward, Kobe 650-0017, Japan; kotanijo0412@gmail.com; 3Emergency and Critical Care Medicine, Tokushima University Hospital, 2-50-1 Kuramoto, Tokushima 770-8503, Japan; manabish777@gmail.com (M.I.); joto@tokushima-u.ac.jp (J.O.)

**Keywords:** lean body mass, body mass index, sepsis, intensive care unit, mortality

## Abstract

Lean body mass is a significant component of survival from sepsis. Several equations can be used for calculating lean body mass based on age, sex, body weight, and height. We hypothesized that lean body mass is a better predictor of outcomes than the body mass index (BMI). This study used a multicenter cohort study database. The inclusion criteria were age ≥18 years and a diagnosis of sepsis or septic shock. BMI was classified into four categories: underweight (<18.5 kg/m^2^), normal (≥18.5–<25 kg/m^2^), overweight (≥25–<30 kg/m^2^), and obese (≥30 kg/m^2^). Four lean body mass equations were used and categorized on the basis of quartiles. The outcome was in-hospital mortality among different BMI and lean body mass groups. Among 85,558 patients, 3916 with sepsis were included in the analysis. Regarding BMI, in-hospital mortality was 36.9%, 29.8%, 26.7%, and 27.9% in patients who were underweight, normal weight, overweight, and obese, respectively (*p* < 0.01). High lean body mass did not show decreased mortality in all four equations. In critically ill patients with sepsis, BMI was a better predictor of in-hospital mortality than the lean body mass equation at intensive care unit (ICU) admission. To precisely predict in-hospital mortality, ICU-specific lean body mass equations are needed.

## 1. Introduction

Muscle is an important component of the body [1]. Muscle contractions enable physical functions, including movements and posture stabilization [2]. In addition to physical functions, muscles play a crucial role in the pumping of blood, temperature management, energy storage, immunological functions, and the production of various cytokines known as myokines [3]. The condition characterized by decreased muscle mass is termed sarcopenia [4]. Recognizing sarcopenia is important because it is associated with the risk of disease progression [5]. Therefore, muscle is considered a vital organ for survival [6,7].

In critically ill patients with sepsis, lean body mass, including muscle mass, is a significant predictor of outcomes [8,9]. A previous study reported that lean body mass at intensive care unit (ICU) admission was associated with survival and physical impairments at discharge [10,11]. Decreased lean body mass reflects a malnutrition status [12]. International clinical nutrition societies recommend the assessment of muscle mass, the main lean body mass component, using the Global Leadership Initiative on Malnutrition (GLIM) criteria for malnutrition assessment [13,14]. Screening is recommended within 24–48 h following ICU admission [15,16]. Therefore, lean body mass assessment is imperative during ICU admission.

Although lean body mass assessment is significant during ICU admission, the method for assessing lean body mass is unclear in critically ill patients. Bioelectrical impedance analysis or dual-energy X-ray absorptiometry can be used for lean body mass assessment [17]. However, in critically ill patients, dynamic fluid changes can affect these assessments [18]. Computed tomography is frequently used for the retrospective analysis of lean body mass assessment; however, it requires transfer to the examination room and exposes patients to radiation. Ultrasonography is an emerging tool for lean body mass assessment. Ultrasonography-based muscle mass assessment is applied in acute [19] or chronic diseases [20]; however, it is not widely used owing to a lack of technical skills. In a previous questionnaire survey, ultrasonography-based muscle mass assessment was conducted only in 14% of health care workers [21].

One type of lean body mass assessment is calculation using equations. As obesity is assessed by calculating the body mass index (BMI), calculating lean body mass from the equation is reasonable. Several studies have reported equations for lean body mass assessment [22,23,24,25]; these equations are significantly correlated with computed tomography-based lean body mass in critically ill patients [26]. As several studies have reported that BMI is a significant predictor of outcomes in critically ill patients, we hypothesized that these lean body mass equations could be more useful for assessing mortality in critically ill patients compared to BMI. In this study, we aimed to compare the calculated lean body mass and BMI for predicting the outcomes of critically ill adults using the Japanese intensive care database.

## 2. Materials and Methods

### 2.1. Study Design

This is a retrospective study using a multicenter cohort study database in the Japanese Intensive Care Patient Database (JIPAD). This study was approved by both the clinical research ethics committees of Tokushima University Hospital (approval number 3721) and the administration office of JIPAD [27]. This study was registered as a clinical trial (UMIN—Clinical Trials Registry: 000039754).

### 2.2. JIPAD

JIPAD is a database of critically ill patients in Japan, established by the Japanese Society of Intensive Care Medicine in 2014. We used the dataset from April 2014 to March 2018. The database included 46 ICUs as of 2018. The quality of the data was maintained by the certification system of the registerer, who passed registration training and quality tests. Furthermore, the quality of the database was periodically examined by the administration office. The dataset was anonymized for analysis.

### 2.3. Patients

Inclusion criteria included (1) age ≥18 years and (2) a diagnosis of sepsis in the primary or secondary disease name record, which included sepsis and septic shock from any infection source. The exclusion criteria involved the presence of missing data and apparently abnormal data for height and body weight required for BMI and lean body mass calculation.

### 2.4. BMI

BMI was calculated by dividing weight (kg) by height squared (m^2^). It was classified into the following four categories based on the World Health Organization: underweight (<18.5 kg/m^2^), normal (≥18.5–<25 kg/m^2^), overweight (≥25–<30 kg/m^2^), and obese (≥30 kg/m^2^) [28].

### 2.5. Lean Body Mass Calculation

Lean body mass was calculated for each patient using the following four separate equations and classified into four categories on the basis of quartiles (lowest, low, high, and highest quartiles).

Equation (1) by Kulkarni et al. [22]:
Males: lean body mass (kg) = −15.605 − (0.032 × age [y]) + (0.192 × height [cm]) + (0.502 × weight [kg])
Females: lean body mass (kg) = −15.034 − (0.018 × age [y]) + (0.165 × height [cm]) + (0.409 × weight [kg])(1)

Equation (2) by Weijs et al. [23]:Lean body mass (kg) = weight (kg) × 0.01 × (100 − [64.5 − 848 × height {m}^2^/weight {kg} + 0.079 × age {y} − 16.4 × sex (1 [male], 0 [female]) + 0.05 × sex (1 [male], 0 [female]) × age (y) + 39.0 × sex (1 [male], 0 [female]) × height [m]^2^/weight [kg])(2)

Equation (3) by Janmahasatian et al. [24]:Males: lean body mass (kg) = (9.27 × 10^3^ × weight [kg])/(6.68 × 10^3^ + 216 × BMI [kg/m^2^])
Females: lean body mass (kg) = (9.27 × 10^3^ × weight [kg])/(8.78 × 10^3^ + 244 × BMI [kg/m^2^])(3)

Equation (4) by Hume et al. [25]:Lean body mass (kg) = 0.32810 × weight (kg) + 0.33929 × height (cm) − 29.5336(4)

### 2.6. Outcome

The outcome was in-hospital mortality. Mortality differences were compared among individuals with underweight, normal weight, overweight, and obesity in BMI, as well as across four quartile categories of lean body mass using the Kulkarni, Weijs, Janmahasatian, and Hume et al. formulas.

### 2.7. Variables

JIPAD data used in the analysis included the following: age; sex; weight; height; comorbidities; date of ICU admission/discharge; ICU admission route, including (1) transfer from the ward, (2) through the emergency room, (3) following elective surgery, and (4) following urgent surgery; primary diagnosis code; secondary diagnosis code; mechanical ventilation; Acute Physiology and Chronic Health Evaluation (APACHE) II and III scores; Simplified Acute Physiology Score (SAPS) II score; and Sequential Organ Failure Assessment (SOFA) score.

### 2.8. Statistical Analysis

Continuous variables were presented as means ± standard deviations or medians (interquartile ranges). Categorical variables were expressed as numbers and percentages. Categorical variables were compared using the χ^2^ test. Bonferroni correction was performed for multiple tests of secondary outcomes. Two-sided *p* values < 0.05 were considered statistically significant. The sample size was not calculated beforehand owing to the exploratory nature of this study. All statistical analyses were performed using Statistical Package for the Social Sciences (version 27, IBM, Armonk, NY, USA).

## 3. Results

### 3.1. Patient Characteristics

During the study period, 85,558 patients were admitted to the ICUs (Figure 1). Among them, 3983 patients met the inclusion criteria. Sixty-seven patients had missing or abnormal data and were excluded. Finally, 3916 samples were included in the analysis. The characteristics of patients are shown in Table 1. The median age of patients was 73 (64–81) years, and 2399 (61.3%) patients were males. Sepsis without and with urinary tract infection were 24.7% and 4.2%, respectively. Septic shock without and with urinary tract infection were 58.9% and 12.1%, respectively. The mortality among the included patients was 1230 (30.7%). The median BMI was 21.6 (19.0–24.5) kg/m^2^. The patient characteristics based on different BMIs and lean body masses are shown in Table 2.

### 3.2. Outcomes

In terms of BMI, the in-hospital mortality rates differed among various weight categories: 36.9%, 29.8%, 26.7%, and 27.9% for underweight, normal weight, overweight, and obesity, respectively (*p* < 0.01, Figure 2). Among them, patients classified as being overweight and normal weight exhibited a significant difference compared with those classified as being underweight (overweight, *p* < 0.05; normal, *p* < 0.05).

In terms of the lean body mass equation, no difference in the in-hospital mortality rate was observed. In the equation by Kulkarni et al., the mortality rates were 32.5%, 29.9%, 31.3%, and 29.2% in quartiles 1, 2, 3, and 4, respectively (*p* = 0.41, Figure 3). In the equation by Weijs et al., the mortality rates were 32.5%, 29.3%, 31.7%, and 29.4% in quartiles 1, 2, 3, and 4, respectively (*p* = 0.32). In the equation by Janmahasatian et al., the mortality rates were 32.7%, 29.0%, 31.6%, and 29.6% in quartiles 1, 2, 3, and 4, respectively (*p* = 0.41). In the equation by Hume et al., the mortality rates were 34.2%, 28.1%, 30.8%, and 29.7% in quartiles 1, 2, 3, and 4, respectively (*p* = 0.03). The morality rates between quartiles 1 and 2 had a significant difference in the equation by Hume et al. (*p* < 0.05).

## 4. Discussion

In this study, we observed that BMI at ICU admission was associated with in-hospital mortality in patients with sepsis. However, contrary to our hypothesis, most calculated lean body mass values were not associated with in-hospital mortality, and all equations did not show lower mortality with increased lean body mass. As lean body mass is reportedly associated with mortality in critically ill patients, the utility of lean body mass calculation was unreliable in this population. BMI was a reliable indicator of in-hospital mortality in critically ill adults with sepsis.

Consistent with a previous meta-analysis [26], BMI was a strong indicator of in-hospital mortality in patients with sepsis. In our study, the obese category had slightly higher mortality than the overweight category, and the obese category did not have a statistical difference in in-hospital mortality. This can be explained by the obesity paradox. Owing to the limited sample size, the obese category included very obese patients (≥40 kg/m^2^). This very obese category reportedly has increased mortality, known as the obesity paradox [29]. As the BMI results were consistent with those of previous studies [30,31,32], our data are reliable.

In a previous study, the four equations used for estimating lean body mass were reportedly correlated with lean body mass quantified using computed tomography (0.680–0.756, *p* < 0.001) [26]. However, they reported that the equation overestimated the lean body mass. This overestimation may have contributed to the nonsignificant outcomes in patients with sepsis. Although a significant difference was partially observed in the equation by Hume et al., the significant difference was inconsistent in the higher quartile population, suggesting unreliable results. Therefore, this equation is considered insufficient for predicting mortality.

To reliably estimate lean body mass, ICU-specific equations are needed. The equation is important not only for predicting mortality but also for assessing nutritional status [13]. Malnutrition is common in critically ill patients, reported to be 38–78% [33] and is associated with poor clinical outcomes, including postoperative complications and mortality [34,35]. As these equations include only age, sex, body weight, and height, reliable equations may require blood tests or anthropometric measurements. These equations will contribute to the prediction of outcomes and the assessment of nutrition status in critically ill patients with sepsis.

Because there are no reliable ICU-specific equations to estimate lean body mass, the establishment of muscle mass assessment may be another strategy for muscle mass assessment at the ICU admission. Bioelectrical impedance analysis, dual-energy X-ray absorptiometry, computed tomography, and ultrasonography are available methods for muscle mass assessment [36]. These methods have pros and cons in terms of their utility [37]. Ultrasonography is recommended because it is noninvasive and not affected by dynamic fluid changes [18]. Although accurate assessment using ultrasonography is influenced by its technical skills, the skill can be acquired through ultrasonography training [38]. Ultrasonography-based muscle mass assessment can be conducted prospectively in the upper [39] or lower limbs [40], which reflect whole-body muscle mass at the ICU admission. Therefore, ultrasonography may become an alternative method for reliable ICU-specific equations.

This study has some limitations. First, the data may include incorrect information. Although data quality was controlled by JIPAD quality control measures, the data may include some incorrect data because we excluded abnormal data. Second, the applicability to different populations requires further studies. The proportion of obesity is different among the populations. Using the Japanese population database, we observed that the BMI is generally lower than that in other countries. Therefore, validity in different populations is required. Third, other potential confounders, including disease severity and comorbidities, may have affected mortality. In this study, we focused on the difference between BMI and calculated lean body mass. Therefore, we did not perform a multivariate analysis. Fourth, this database distinguished between sepsis, sepsis along with a urinary tract infection, septic shock, and septic shock along with a urinary tract infection. Therefore, we could not identify the cause of sepsis except for a urinary tract infection. Because sepsis is caused by various sources, further studies are required on the different infection sources.

## 5. Conclusions

In critically ill patients with sepsis, BMI was a better predictor than calculated lean body mass at ICU admission. To reliably estimate lean body mass and mortality, ICU-specific equations are needed.

## Figures and Tables

**Figure 1 diseases-12-00030-f001:**
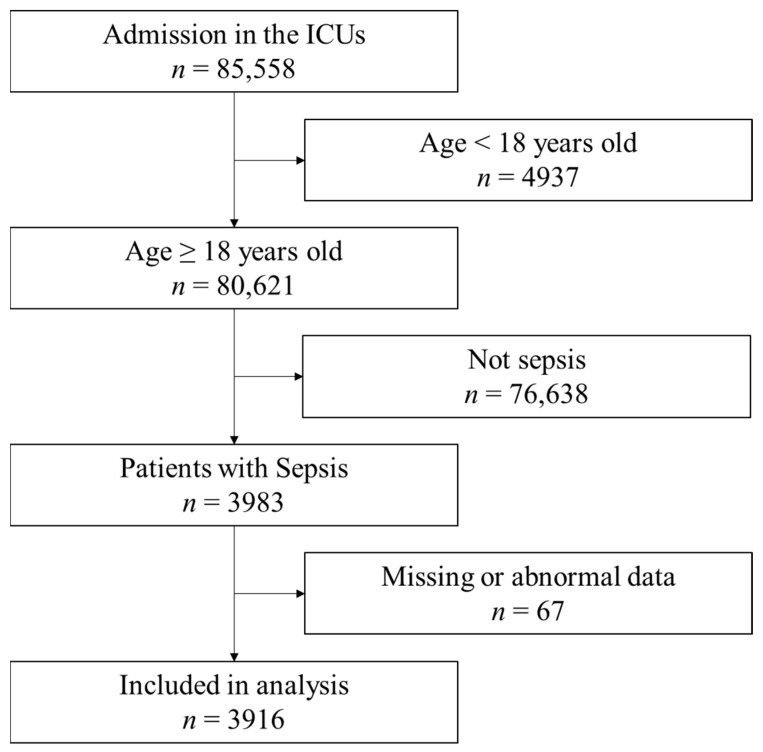
Flow of data selection from the Japanese Intensive Care Patient Database. Out of the 85,558 registry entries, the analysis included 3,916 patients. ICU: intensive care unit.

**Figure 2 diseases-12-00030-f002:**
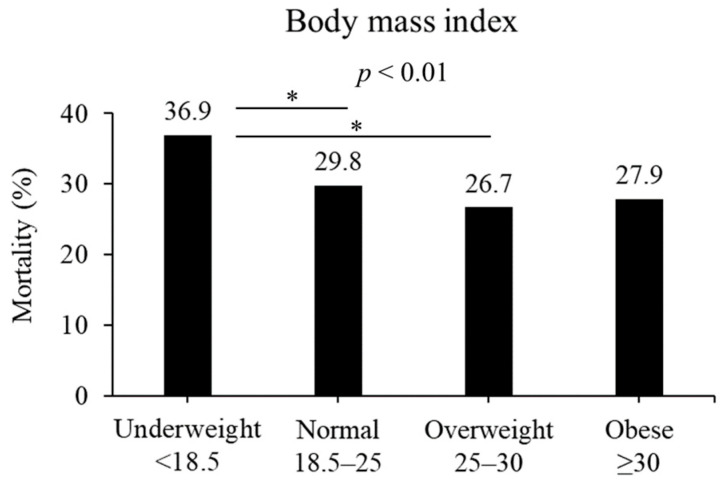
Mortality based on BMI (kg/m^2^). Mortality was different in the four BMI groups (*p* < 0.01). Underweight patients had significantly higher mortality than normal or overweight in post hoc analysis. * *p* < 0.05 in post hoc Bonferroni tests.

**Figure 3 diseases-12-00030-f003:**
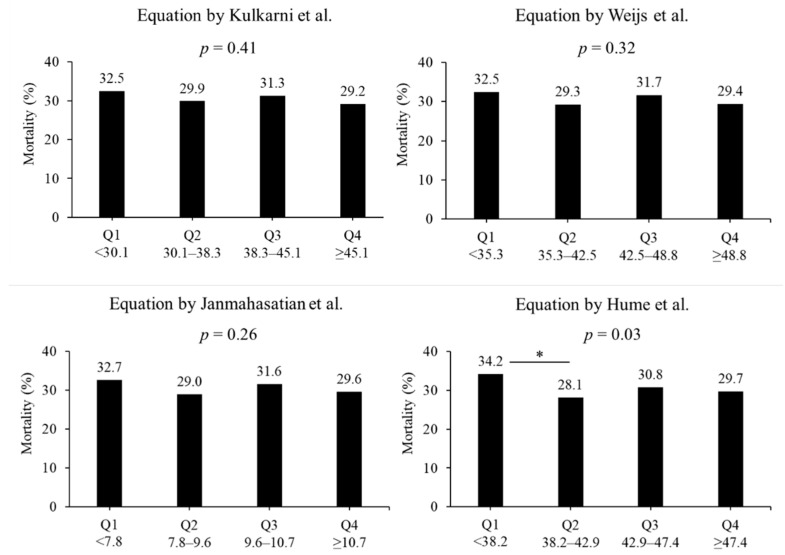
Mortality based on four lean body mass equations. Four equations include the formula reported by Kulkarni et al. [22], Weijs et al. [23], Janmahasatian et al. [24], and Hume et al. [25] Groups were divided into lowest quartile, Q1, to highest quartile, Q4. * *p* < 0.05 in post hoc Bonferroni tests. Q: quartile.

**Table 1 diseases-12-00030-t001:** Characteristics of Patients.

Variables	Overall (*n* = 3916)
Age, mean ± SD, y	70.5 ± 14.1
Male, *n* (%)	2399 (61.3%)
Weight	55.0 (46.8–64.2)
Height	160.0 (152.0–167.0)
Comorbidities, *n* (%)	
Chronic heart failure	58 (1.5%)
Chronic respiratory failure	105 (2.7%)
Chronic liver failure	36 (0.9%)
Immunocompromised	547 (14.0%)
Sepsis classifications, *n* (%)	
Sepsis	968 (24.7%)
Sepsis with urinary tract infection	166 (4.2%)
Septic shock	2307 (58.9%)
Septic shock with urinary tract infection	475 (12.1%)
Mechanical ventilation, *n* (%)	1222 (31.2%)
APACHE II score	23 (18–30)
APACHE III score	89 (71–114)
SAPS II score	51 (39–66)
SOFA score	8 (6–12)
ICU admission route, *n* (%)	
Transfer from the ward	1459 (37.3%)
Through the emergency room	1447 (37.0%)
Following elective surgery	747 (19.1%)
Following urgent surgery	69 (1.8%)
Length of ICU stay	7 (3–19)
Mortality, *n* (%)	1203 (30.7%)
Body mass index (kg/m^2^)	21.6 (19.0–24.5)
Underweight (<18.5 kg/m^2^), *n* (%)	838 (21.4%)
Normal (≥18.5 to <25 kg/m^2^), *n* (%)	2229 (56.9%)
Overweight (≥25 to <30 kg/m^2^), *n* (%)	652 (16.6%)
Obese (≥30 kg/m^2^), *n* (%)	197 (5.0%)
Lean body mass by Kulkarni et al. [22], kg	38.3 (30.1–45.1)
Lean body mass by Weijs et al. [23], kg	42.5 (35.3–48.8)
Lean body mass by Janmahasatian et al. [24], kg	42.6 (33.8–49.5)
Lean body mass by Hume et al. [25], kg	42.9 (38.2–47.4)

SD: standard deviation; APACHE: Acute Physiology and Chronic Health Evaluation; SAPS: Simplified Acute Physiology Score; SOFA: Sequential Organ Failure Assessment; ICU: intensive care unit. Data are presented as median (IQR) unless otherwise indicated.

**Table 2 diseases-12-00030-t002:** Characteristics of patients with different BMIs and lean body masses.

BMI	Underweight	Normal	Overweight	Obese
Age, mean ± SD, y	70.8 ± 15.8	71.4 ± 13.2	69.6 ± 13.4	62.5 ± 14.8
Male, *n* (%)	471 (56.2%)	1422 (63.8%)	409 (62.7%)	97 (49.2%)
APACHE II score	24 (19–31)	23 (18–30)	22 (17–29)	23 (18–32)
SOFA score	8 (5–11)	8 (6–11)	9 (6–12)	10 (7–13)
Lean body mass by Kulkarni et al.	Q1	Q2	Q3	Q4
Age, mean ± SD, y	74.3 ± 14.7	72.1 ± 13.5	70.4 ± 12.9	65.3 ± 13.5
Male, *n* (%)	47 (4.8%)	487 (49.7%)	910 (93.0%)	955 (97.5%)
APACHE II score	24 (18–30)	24 (18–30)	23 (18–30)	23 (17–30)
SOFA score	8 (5–11)	9 (6–12)	8 (5–11)	9 (6–12)
Lean body mass by Weijs et al.	Q1	Q2	Q3	Q4
Age, mean ± SD, y	75.7 ± 13.6	71.7 ± 13.9	70.5 ± 12.5	64.1 ± 13.7
Male, *n* (%)	58 (5.9%)	474 (48.4%)	906 (92.5%)	961 (98.2%)
APACHE II score	24 (19–30)	23 (18–30)	23 (18–30)	23 (17–29)
SOFA score	8 (5–12)	9 (6–11)	8 (5–11)	9 (6–12)
Lean body mass by Janmahasatian et al.	Q1	Q2	Q3	Q4
Age, mean ± SD, y	73.8 ± 15.1	71.6 ± 13.6	70.6 ± 13.0	66.1 ± 13.3
Male, *n* (%)	41 (4.2%)	457 (46.7%)	927 (94.7%)	974 (99.5%)
APACHE II score	24 (18–30)	23 (18–30)	24 (18–30)	22 (17–30)
SOFA score	8 (5–11)	9 (6–11)	8 (5–11)	9 (6–12)
Lean body mass by Hume et al.	Q1	Q2	Q3	Q4
Age, mean ± SD, y	75.7 ± 13.7	71.8 ± 13.5	70.1 ± 12.7	64.4 ± 13.9
Male, *n* (%)	169 (17.3%)	524 (53.5%)	805 (82.2%)	901 (92.0%)
APACHE II score	24 (19–30)	23 (18–30)	23 (17–30)	23 (17–30)
SOFA score	8 (5–11)	9 (6–11)	8 (5–11)	9 (6–12)

SD: standard deviation; APACHE: Acute Physiology and Chronic Health Evaluation; SOFA: Sequential Organ Failure Assessment. Data are presented as median (IQR) unless otherwise indicated.

## Data Availability

The datasets used in this study are not available since the dataset was obtained from the Japanese Intensive Care Patient Database through a formal request/approval process.

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
