# Peer review of "Utility of Lean Body Mass Equations and Body Mass Index for Predicting Outcomes in Critically Ill Adults with Sepsis: A Retrospective Study"

_diseases, 2024, doi:10.3390/diseases12020030_

Round 1

Reviewer 1 Report

Comments and Suggestions for Authors

The authors have developed a well-conducted and well-written study with the aim of comparing calculated lean body mass and BMI to predict outcomes of critically ill adults using the Japanese intensive care database.

However, I would like to make a few observations before recommending their work for publication.

1. Please, could the authors detail in the title that your study is retrospective?

2. Please could the authors add a figure footnote to Figures 1-3?

3. It is necessary to add the formula of the sample size calculation used

4. The Introduction and Discussion section is very brief...this in some manuscripts may be an advantage...but I advise the authors to briefly detail in the Introduction section also the increasing use of ultrasound in rehabilitation to assess muscle contraction, detect possible tendinopathies, as well as to serve as a guide in the treatment, recommending to comment on the following high quality papers: DOI: 10.3390/diagnostics11040632

Comments on the Quality of English Language

No comments

Author Response

Responses to Reviewer #1:

  1. The authors have developed a well-conducted and well-written study with the aim of comparing calculated lean body mass and BMI to predict outcomes of critically ill adults using the Japanese intensive care database.

answer: We appreciate reviewer’s positive comments.

  1. Please, could the authors detail in the title that your study is retrospective?

answer: We added “retrospective” in the title as following.

“Utility of Lean Body Mass Equations and Body Mass Index for Predicting Outcomes in Critically Ill Adults With Sepsis: A Retrospective Study”

  1. Please could the authors add a figure footnote to Figures 1-3?

answer: We sincerely added the figure footnote to Figure 1-3.

  1. It is necessary to add the formula of the sample size calculation used?    

answer: Since this is an exploratory study, we could not calculate the sample size. We added the reason why we included this sample size as following.

“Sample size was not calculated a priory due to the exploratory nature of this study.”

  1. The Introduction and Discussion section is very brief...this in some manuscripts may be an advantage...but I advise the authors to briefly detail in the Introduction section also the increasing use of ultrasound in rehabilitation to assess muscle contraction, detect possible tendinopathies, as well as to serve as a guide in the treatment, recommending to comment on the following high quality papers: DOI: 10.3390/diagnostics11040632?    

answer: We appreciate the suggestion of the high quality paper. We cited the article. We added one more paragraph for more detailed information in both of background and discussion.

Reviewer 2 Report

Comments and Suggestions for Authors

The authors intended to investigate the role of BMI for ICU patient with sepsis. This is a comparative simple study without any scientific value. I would recommend rejection due to lack of scientific soundness. 

1. The authors failed to correlate the inflammation related cytokines and chemokines with BMI and sepsis.

2. The authors failed to differentiate the infectious bacteria for this study.

3. The authors even failed to present figure legends for each figure. This simple mistakes should not be happened before manuscript submission.

4. The basic diseases of these patients should be included.

5. As the authors subdivided the patients into underweight, normal, overweight and obese, the basic characteristics should be subdivided to these four groups as well.

Comments on the Quality of English Language

The manuscript is poorly written with lots of grammar mistakes and typos.

Author Response

Responses to Reviewer #2:

  1. The authors intended to investigate the role of BMI for ICU patient with sepsis. This is a comparative simple study without any scientific value. I would recommend rejection due to lack of scientific soundness.

Answer: We appreciate the time and effort spent by the reviewer. We revised to live up to your expectations for this brief report.

  1. The authors failed to correlate the inflammation related cytokines and chemokines with BMI and sepsis.

Answer: Thank you for the opportunity to clarify this point. As mentioned by the reviewer, the inflammation-related cytokines and chemokines are very important in sepsis. It is meaningful to analyze the correlation between inflammation and sepsis. However, this is a retrospective study analyzing the Japanese Intensive care PAtient Database (JIPAD). This database did not include these cytokine data. We would like to pursue this relationship in the future study.

  1. The authors failed to differentiate the infectious bacteria for this study.

Answer: This is also important point. The infectious bacteria are important for analysis. However, this is the retrospective study using JIPAD database, which did not include the information of infectious bacteria. This manuscript is a brief report, and we are planning to conduct future studies including cytokines and infectious bacteria information after this brief report.

  1. The authors even failed to present figure legends for each figure. This simple mistake should not be happened before manuscript submission.

Answer: We sincerely added the figure footnote to Figure 1-3.

  1. The basic diseases of these patients should be included.

Answer: We added the detail of sepsis in the table 1. In this database, we can only classify into four categories: sepsis, sepsis with urinary tract infection, septic shock, and septic shock with urinary tract infection. This is the limitation of this retrospective study. We cannot analyze further sources of sepsis. We added this point as following. 

Manuscript

“Sepsis without and with urinary tract infection were 24.7% and 4.2%, respectively. Septic shock without and with urinary tract infection were 58.9% and 12.1%, respectively.”

Limitation

“Fourth, this database distinguished between sepsis, sepsis along with urinary tract infection, septic shock, and septic shock along with urinary tract infection. Therefore, we could not identify the cause of sepsis except for a urinary tract infection. Because sepsis is caused by various sources, further studies are required on the different infection sources.”

  1. As the authors subdivided the patients into underweight, normal, overweight and obese, the basic characteristics should be subdivided to these four groups as well.

Answer: We added the basic characteristics in underweight, normal, overweight and obese as table 2.

  1. The manuscript is poorly written with lots of grammar mistakes and typos.

Answer: We checked the grammar mistakes and typos throughout the manuscript, and revised these mistakes. Furthermore, we had English editing service by ENAGO English editing company.

Round 2

Reviewer 1 Report

Comments and Suggestions for Authors

The current version of the authors' manuscript has improved, so I recommend its publication.

Congratulations

Comments on the Quality of English Language

No comments

Reviewer 2 Report

Comments and Suggestions for Authors

The authors carefully revised the manuscript according to the reviewer's suggestion.

Comments on the Quality of English Language

The manuscript should be extensively revised by native English speakers.